# Combination of EGFR-Directed Tyrosine Kinase Inhibitors (EGFR-TKI) with Radiotherapy in Brain Metastases from Non-Small Cell Lung Cancer: A 2010–2019 Retrospective Cohort Study

**DOI:** 10.3390/cancers15113015

**Published:** 2023-06-01

**Authors:** Vineeth Tatineni, Patrick J. O’Shea, Shreya Saxena, Atulya A. Khosla, Ahmad Ozair, Rupesh R. Kotecha, Xuefei Jia, Yasmeen Rauf, Erin S. Murphy, Samuel T. Chao, John H. Suh, David M. Peereboom, Manmeet S. Ahluwalia

**Affiliations:** 1Rosa Ella Burkhart Brain Tumor and Neuro-Oncology Center, Taussig Cancer Institute, Cleveland Clinic, Cleveland, OH 44106, USA; 2Case Western Reserve University School of Medicine, Cleveland, OH 44106, USA; 3Miami Cancer Institute, Baptist Health South Florida, Miami, FL 33176, USAahmad.ozair@baptisthealth.net (A.O.);; 4Department of Medical Oncology, Taussig Cancer Institute, Cleveland Clinic, Cleveland, OH 44106, USA; 5Division of Neuro-Oncology, University of North Carolina, Chapel Hill, NC 27599, USA; 6Department of Radiation Oncology, Taussig Cancer Institute, Cleveland Clinic, Cleveland, OH 44106, USA; 7Herbert Wertheim College of Medicine, Florida International University, Miami, FL 33199, USA

**Keywords:** epidermal growth factor receptor, brain metastasis, lung carcinoma, targeted therapy, molecular therapy, whole-brain radiation therapy, stereotactic radiosurgery

## Abstract

**Simple Summary:**

Radiotherapy, in the form of either whole-brain radiotherapy (WBRT) or stereotactic radiosurgery (SRS), continues as the standard of care for patients of non-small cell lung cancer with brain metastases (NSCLCBM). Recently, targeted therapies have emerged as systemic options for brain metastases with certain genetic mutations. Tyrosine kinase inhibitors directed against EGFR protein (EGFR-TKI) have come forth as the preferred treatment of EGFR-mutated NSCLC and have also shown promise in NSCLCBM. However, there have been few studies comparing the synergistic effects of EGFR-TKIs and radiotherapy in NSCLCBM. This study is one of the few that investigates survival rates between standard radiotherapy modalities and a combination of radiotherapies and EGFR-TKIs. Our data may help guide clinicians in future treatment plans for EGFR-mutated NSCLCBM patients.

**Abstract:**

Introduction: Traditionally, brain metastases have been treated with stereotactic radiosurgery (SRS), whole-brain radiation (WBRT), and/or surgical resection. Non-small cell lung cancers (NSCLC), over half of which carry EGFR mutations, are the leading cause of brain metastases. EGFR-directed tyrosine kinase inhibitors (TKI) have shown promise in NSCLC; but their utility in NSCLC brain metastases (NSCLCBM) remains unclear. This work sought to investigate whether combining EGFR-TKI with WBRT and/or SRS improves overall survival (OS) in NSCLCBM. Methods: A retrospective review of NSCLCBM patients diagnosed during 2010–2019 at a tertiary-care US center was performed and reported following the ‘strengthening the reporting of observational studies in epidemiology’ (STROBE) guidelines. Data regarding socio-demographic and histopathological characteristics, molecular attributes, treatment strategies, and clinical outcomes were collected. Concurrent therapy was defined as the combination of EGFR-TKI and radiotherapy given within 28 days of each other. Results: A total of 239 patients with EGFR mutations were included. Of these, 32 patients had been treated with WBRT only, 51 patients received SRS only, 36 patients received SRS and WBRT only, 18 were given EGFR-TKI and SRS, and 29 were given EGFR-TKI and WBRT. Median OS for the WBRT-only group was 3.23 months, for SRS + WBRT it was 3.17 months, for EGFR-TKI + WBRT 15.50 months, for SRS only 21.73 months, and for EGFR-TKI + SRS 23.63 months. Multivariable analysis demonstrated significantly higher OS in the SRS-only group (HR = 0.38, 95% CI 0.17–0.84, *p* = 0.017) compared to the WBRT reference group. There were no significant differences in overall survival for the SRS + WBRT combination cohort (HR = 1.30, 95% CI = 0.60, 2.82, *p* = 0.50), EGFR-TKIs and WBRT combination cohort (HR = 0.93, 95% CI = 0.41, 2.08, *p* = 0.85), or the EGFR-TKI + SRS cohort (HR = 0.46, 95% CI = 0.20, 1.09, *p* = 0.07). Conclusions: NSCLCBM patients treated with SRS had a significantly higher OS compared to patients treated with WBRT-only. While sample-size limitations and investigator-associated selection bias may limit the generalizability of these results, phase II/III clinicals trials are warranted to investigate synergistic efficacy of EGFR-TKI and SRS.

## 1. Introduction

Lung cancer is the second most common type of cancer worldwide, with non-small cell lung cancer (NSCLC) being the most common subtype and the most common cause of brain metastases [1,2]. In addition to lung cancer itself being a leading cause of cancer mortality, the development of brain metastases adds considerable symptoms, a poorer prognosis, and a much poorer quality of life [3,4]. Management of brain metastases with systemic therapies, particularly chemotherapies, has been challenging due to poor blood–brain-barrier penetration, leading to a low intracranial response rate [4]. Therefore, brain metastases have been historically treated with stereotactic radiosurgery (SRS), whole-brain radiation (WBRT), surgical resection, or a combination of these treatments [4].

In recent years, the management of NSCLCBM has shifted from traditional radiation and surgical therapy to targeted molecular therapies [5]. While several molecular agents have been studied, only a few have demonstrated meaningful utility as either therapeutic targets or prognostic markers [5]. Epidermal growth factor receptor (EGFR), a transmembrane growth factor receptor tyrosine kinase, is mutated in 40–60% of NSCLCs [6]. The risk of developing brain metastases is higher in patients with EGFR mutations, though, fortunately, EGFR signaling pathways have also become an effective targetable marker [7].

EGFR tyrosine kinase inhibitors (EGFR-TKIs) were first introduced in the early 2000s and have been proven to be more effective than standard chemotherapy [8,9]. First-generation EGFR-TKIs are limited by their ability to cross the blood–brain barrier (BBB) and are ineffective against certain tumor mutations [10,11]. This, of course, limits their ability to treat an EGFR-mutated lung tumor that has metastasized to the brain. Second-generation EGFR-TKIs have improved activity against exon 19 deletion mutations and therefore have better efficacy than first-generation agents [11,12,13]. Third-generation EGFR-TKIs have shown both better BBB penetration and efficacy against T790M mutations [13], making them, in theory, the most effective against metastatic lung tumors in the brain.

Though these new targeted therapies have shown great promise in NSCLC, there are few robust data when looking specifically at NSCLCBM [14,15]. Fan et al. conducted a systemic review of 16 clinical studies where the pooled analysis indicated EGFR-TKIs are effective for patients with NSCLCBM [16].

The development of resistance to EGFR-TKIs is inevitable, and a few trials have analyzed the efficacy of its combination with WBRT and SRS in patients with EGFR-mutated NSCLCBM [17,18]. In parallel to the increasing utility of EGFR-TKIs, stereotactic radiosurgery (SRS) has also emerged as a superior tool for radiotherapy in brain metastases, compared to whole-brain teletherapy approaches [19,20]. SRS is now commonly used as first-line local therapy for brain metastases [21]. The interaction of EGFR-TKI therapy and SRS together has been a topic of interest in recent years [22,23,24]. We look to add to this field of newer interest with our large, single-institution study.

As EGFR-TKI data in NSCLCBM becomes more comprehensive, synergistic therapies need to be looked at. There is no consensus on its management, and the efficacy of the combination regimen of EGFR-TKIs and radiotherapy remains unclear among patients with various mutation subtypes. Therefore, we aimed to evaluate the OS in NSCLCBM patients treated with WBRT only, SRS only, WBRT and SRS, and a combination of EGFR-TKIs and SRS and EGFR-TKIs and WBRT.

## 2. Methods

### 2.1. Study Design, Patient Population, and Selection

A multi-arm retrospective cohort study was conducted, after institutional review board (IRB) approval, and reported following the ‘strengthening the reporting of observational studies in epidemiology’ (STROBE) guidelines. We investigated all EGFR-mutated NSCLCBM patients diagnosed during 2010–2019 at Cleveland Clinic, Ohio, a tertiary-care institution in the US.

We included all patients >18 years of age with EGFR-mutated NSCLCBM who were treated with SRS, WBRT, or EGFR-TKIs, as first-line therapy after the diagnosis of brain metastases. We did not exclude patients who received intracranial surgery at any point in the disease course. We also did not exclude patients who received EGFR-TKIs or other systemic therapies before the diagnosis of brain metastases as long as there was a change in therapy after the diagnosis of brain metastases since our primary goal was to evaluate the OS rates after the diagnosis of brain metastases. Patients who were included in our study were followed in the outpatient setting approximately every 3 months. Overall survival (OS) was defined as the date of first therapy after the diagnosis of brain metastases until the date of the last progress note or date of death.

Patient characteristics, initial imaging, genomic analysis, and treatment details were collected from the institution’s electronic medical records. Data were recorded in a secure online database and then exported for statistical analysis. Characteristic information collected includes the Karnofsky Performance Score (KPS), age, race, and sex. The treatment details collected include the date of therapy initiation and the line of therapy.

### 2.2. EGFR-TKI Data

Erlotinib, gefitinib, afatinib, and osimertinib were primarily investigated in this study, whereas dacomitinib was not evaluated as no patients received this drug in our cohort. There was no EGFR-TKI-only cohort because all patients who received EGFR-TKIs also received some form of radiation during their treatment course. It is possible that patients in the EGFR-TKIs cohorts received more than one line of EGFR-TKI throughout their treatment course after the diagnosis of brain metastases; however, OS was only calculated based on the initiation of EGFR-TKI therapy after the diagnosis of brain metastases.

### 2.3. SRS Data

Patients in the SRS-only cohort were treated with SRS as first-line therapy and did not receive WBRT or systemic therapies after diagnosing brain metastases. However, the patients may have received further treatments for SRS throughout their disease course. The number of lesions that were treated upfront was not subclassified. The combination cohort of EGFR-TKIs and SRS was used to investigate a synergistic treatment approach. Concurrence was defined as therapies given within 28 days of each other.

### 2.4. WBRT Data

Patients in the WBRT-only cohort were treated with WBRT as first-line therapy and did not receive SRS or any systemic therapies after diagnosing brain metastases. Patients in the WBRT-only cohort received one course of WBRT; no repeat treatment courses were given. The date of WBRT was defined as the first date of radiation treatment given to the patient over the full course of treatment. Concurrence for the combination EGFR-TKI + WBRT cohort was defined as treatments given within 28 days of each other. Nonconcurrent treatments were not investigated. The specific radiation dosage, length of WBRT course, or discontinuation of treatment due to symptoms were not subclassified. The WBRT-only cohort was the reference cohort due to its history as the traditional modality of treatment of brain metastases.

### 2.5. SRS + WBRT Data

Patients in the SRS + WBRT were treated concurrently with both modalities within 28 days of each other. Patients in this cohort were not treated with any form of systemic therapy after the diagnosis of brain metastases. This cohort may have received either SRS or WBRT further along in the disease treatment course. We again did not subclassify the number of lesions treated initially with SRS, the length of the WBRT course, or discontinuation of WBRT due to symptoms.

### 2.6. Data Analysis

Categorical clinicopathologic factors were summarized as frequency counts and percentages, and continuous factors as medians and ranges. OS was measured from the start date of the first treatment received to the date of the last follow-up or date of death and was summarized using the Kaplan–Meier method. The 1-year and 2-year survival rates and estimated median survival for each treatment cohort were reported. The Cox proportional hazard model with a two-sided Wald test was used to evaluate the impact of the treatment on OS. The survival model was adjusted by clinical variables selected using the random forest method. The primary model was adjusted by the variables mostly identified as prognostic factors in patients with NSCLCBM in previous studies [25]. These variables were age at diagnosis of brain metastases, gender, number of brain metastases, the existence of extracranial metastases, the existence of leptomeningeal metastases, KPS, symptomatic at time of brain metastases, and the duration from the date of diagnosis of brain metastases to the date of treatment.

Due to its historical use and previously being the standard of care, the WBRT-only cohort was used as the reference cohort to which we compared OS. Progression-free survival (PFS) was not calculated in our study due to the lack of specific magnetic resonance imaging (MRI) intracranial-lesion data and difficulty with consistent, unbiased alternative definitions of progression. Statistical significance was defined as a *p*-value (*p*) of <0.05. All statistical analyses were performed using R Statistical Software version 4.1.0 (R Foundation for Statistical Computing, Vienna, Austria).

## 3. Results

### 3.1. Patient Characteristics

Between 2010 and 2019, our retrospective study found a total of 239 patients who had NSCLCBM with EGFR mutations. Of these, a total of 32 patients received WBRT alone, another 51 patients received SRS alone, 36 patients were treated with SRS + WBRT combined, 29 patients received EGFR-TKI + WBRT, and 18 patients received combination EGFR-TKI + SRS.

The WBRT-only group had a median age of 68.4 years, with 62.5% being female. The SRS-only group had a median age of 62.7 years, with 70.6% female, while the SRS + WBRT cohort had a median age of 64.8 years, with 55.6% being female. The combination EGFR-TKI + SRS cohort had a median age of 70.5 years, with 50% being females, and the combination EGFR-TKI + WBRT cohort had a median age of 61.5 years with 58.6% being females. Further characteristics of the five subdivided cohorts are shown in Table 1.

There were statistically significant differences in the proportion of patients with single versus multiple brain metastases, the proportion of patients with versus without extracranial metastases, and the proportion of cases with and without leptomeningeal spread in each cohort.

Multivariable analysis permitted the selection of three key adjustment variables, all of which were significantly associated with poorer survival: KPS < 70 (*p* = 0.002), age (*p* = 0.042), and time from brain-metastases diagnosis to initiation of treatment (*p* = 0.005).

### 3.2. Single Therapies

The estimated median OS for the WBRT-only cohort was 3.23 months, with a 1-year OS rate of 35% (95% confidence interval (CI) = 17%, 54%) and a 2-year OS rate of 25% (95% CI = 10%, 44%). The estimated median OS for the SRS-only cohort was 21.73 months, with a 1-year OS rate of 68% (95% CI = 51%, 80%) and a 2-year OS rate of 39% (95% CI = 22%, 55%). Under multivariable analysis, when using the WBRT cohort as the reference cohort, the hazard ratio of the SRS-only group was 0.33 (95% CI = 0.17, 0.62), showing a statistically significant difference between the WBRT-only and the SRS-only cohorts (*p* < 0.001) (Table 2, Figure 1).

### 3.3. Combination Therapies

The estimated median OS for the WBRT + SRS combination cohort was 3.17 months, with a 1-year OS rate of 26% (95% CI = 12%, 42%) and a 2-year OS rate of 6% (95% CI = 1%, 19%).

The median OS for the EGFR-TKI + WBRT cohort was 15.50 months, and for the EGFR-TKI + SRS cohort was 23.63 months. The proportion of patients with 1-year OS for the two cohorts was 64% (95%CI = 42%, 79%) and 71% (95% CI = 43%, 87%), respectively. The 2-year OS rate was 28% (95% CI = 12%, 46%) and 37% (95% CI = 12%, 62%), respectively.

Through a multivariable analysis using the WBRT-only cohort as a reference, we found no significant difference in overall survival for the SRS + WBRT combination cohort (HR = 1.30, 95% CI = 0.60, 2.82, *p* = 0.50), the EGFR-TKIs and WBRT combination cohort (HR = 0.93, 95% CI = 0.41, 2.08, *p* = 0.85), and the EGFR-TKI + SRS cohort (HR = 0.46, 95% CI = 0.20, 1.09, *p* = 0.07). However, overall survival was much higher in the multivariate analysis for the SRS-only cohort (HR = 0.38, 95% CI = 0.17, 0.84, *p* = 0.017) (Table 3).

## 4. Discussion

### 4.1. Relevance to Literature and Clinical Practice

In this large database of EGFR-mutated NSCLCBM patients treated at a tertiary-care center, SRS alone was found to have significantly better overall survival compared to WBRT alone, and to the SRS + WBRT, SRS + TKI cohorts, and TKI + WBRT cohorts. These findings expand on the previous studies that looked at the use of TKIs and radiotherapy in EGFR-mutated NSCLCBM [22,23,24]. Jia et al. compared the efficacy of combination of TKI with SRS or WBRT in NSCLCBM patients and reported an increased survival in the TKI + SRS cohort (25.1 months vs. 22.0 months, *p* = 0.042) [26].

Magnuson et al. specifically found that SRS combined with EGFR-TKI for upfront treatment resulted in the longest OS. However, our study compared combination therapies to WBRT only instead of combination therapies to a cohort of EGFR-only patients. Cheng et al. demonstrated increased survival in patients with NSCLCBM treated with SRS + 1st-gen TKI compared to TKI alone, and to addition of sequential 3rd-gen TKI: osimertinib further prolonged survival (43.5 vs. 24.3 months, *p* < 0.001) [27]. Similar to our results, WBRT in combination with TKI did not improve OS compared to WBRT alone (12.9 vs. 10.0 months, *p* = 0.5) in a multicenter phase III trial [28]. Zhai et al. also showed that the combination of WBRT and TKI did not increase the overall survival, but for the subset of patients with the EGFR L858R mutation, where the combination led to better survival (*p* = 0.046) [29]. A similar study by He et al. showed no improvement in OS in the cohort receiving the combination of WBRT and TKI compared to TKI alone, but the combination did improve intracranial PFS among patients with more than three brain metastases from NSCLC [30]. Finally, Chiou et al. reported improved intracranial tumor control rates in patients with NSCLCBM receiving a combination of TKI + SRS compared to TKI alone (79.8% vs. 31.2%, *p* < 0.0001) [31].

The BRAIN study was a phase III randomized clinical trial which compared the efficacy of icotinib, a first-gen EGFR-TKI with WBRT to treat NSCLCBM. After a median follow-up of 16.5 months, the intracranial PFS (iPFS) with icotinib was found to be 10.0 months vs. 4.8 months with WBRT, equating to a 44% reduction in event of intracranial progression (*p* = 0.014) [32]. In a retrospective study by Fan et al., icotinib was combined with RT in patients with NSCLCBM, and an improvement in iPFS was observed compared to the cohort receiving icotinib alone (22.4 vs. 13.9 months respectively, *p* = 0.043) [33]. Similar results were observed in studies utilizing erlotinib and gefitinib as first-generation EGFR-TKIs, which, when combined with RT, demonstrated superior outcomes than EGFR-TKIs alone [34,35].

Our study differs from previously mentioned studies as it also investigated an isolated SRS cohort, as opposed to radiotherapy more generally. Though our data showed the longest estimated OS for the combination of SRS and EGFR-TKIs, the SRS-only cohort also showed a statistical significance in OS rate. When available, SRS has become the prevailing first-line radiation for brain metastases; however, our data do not definitively conclude that the combination of EGFR-TKI and SRS therapy is superior to the SRS-only cohort [19,20,21].

With controlled variables, the current findings suggest that the improvement in OS is due to the improvement of CNS disease rather than selection bias or difference between patient cohorts. These findings advocate that a combination treatment of SRS + EGFR-TKIs may perhaps be the future standard of care for EGFR-mutated NSCLCBM. Though our data do not prove the superiority of combination treatment over SRS only, these results are likely the result of an underpowered cohort, and clinicians should consider using a combination approach with targeted therapy.

EGFR-TKIs provide a systemic approach, which has been proven to help control NSCLC and extracranial metastases [14,36]. Studies have also shown that EGFR-mutated NSCLC is highly sensitive to radiation, which can explain the benefit of using upfront SRS in these patients [37,38]. SRS has demonstrated high local control rates in EGFR-mutated brain metastases, with increasing utilization in this cohort [37,39]. SRS has also been shown with frequency to be an improved option over WBRT, with noninferior survival and decreased toxicity [24,40,41]. Kim et al. also similarly found no improvement in PFS between concurrent SRS + TKI and SRS alone; however, they noticed no additional adverse events related to concurrent therapy [42].

### 4.2. Limitations

Several limitations limit the generalizability of the current work. First, this was a retrospective study with limited adjustment for confounders, particularly the unknown ones. Second, given the observational setting, investigator-associated selection bias may have led to different group characteristics which likely impacted the results. Although certain variables were controlled for, patients were chosen for their specific treatment for several reasons, and these could have a confounding impact on survival. Third, this study was performed at a large academic tertiary-care center, and all places of care may not have access to the same treatments.

Finally, our study only evaluated OS; we were not able to evaluate PFS due to the lack of recorded MRI images in our database and difficulty with accurately evaluating progression without intracranial imaging. Without evaluating PFS, we were also unable to have an EGFR-TKI-alone cohort since all patients who received EGFR-TKI upfront also received radiotherapy at some point in their disease course, which may cause a confounding effect on estimated OS.

### 4.3. Potential Directions

For future studies, we believe an EGFR-TKI-alone cohort should be used along with the exploration of PFS based on the modified ‘Response Assessment for Neuro-Oncology’ (RANO) criteria for brain metastases [43,44]. This would provide more accurate and helpful data on whether there truly is a synergistic component to combining EGFR-TKIs and SRS treatments. Our work did not distinguish which treatment was given first and defined concurrence as within 28 days of each other. Future studies may consider utilizing randomized controlled trial design, or large prospective cohort studies should be used to investigate concurrent SRS and TKIs vs. SRS followed by TKIs vs. SRS alone.

Notably, a multi-institution retrospective study showed SRS followed by TKI was superior to TKI followed by SRS [24]. Another retrospective brain metastasis study showed minimal difference in OS between concurrent TKI and SRS treatment compared to SRS followed by TKI, but with a relatively small sample [45]. Until such investigations are performed to elucidate whether concurrent TKI/SRS is superior to SRS followed by TKI, upfront SRS may be considered the standard of care, with TKI use at clinical discretion.

## 5. Conclusions

This study demonstrates a significantly higher survival for NSCLCBM patients treated with SRS alone compared to patients treated with WBRT only. While sample-size limitations and investigator-selection bias may limit the validity of these results, phase II/III randomized controlled trials are warranted to demonstrate this synergistic effect through high-quality evidence.

## Figures and Tables

**Figure 1 cancers-15-03015-f001:**
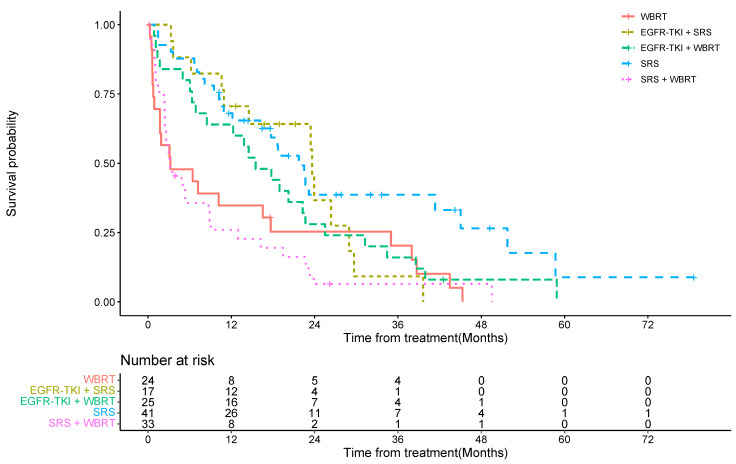
Kaplan–Meier curves demonstrating the overall survival in various cohorts. WBRT: whole-brain radiation therapy, SRS: stereotactic radiosurgery, EGFR: epithelial growth factor receptor, TKI: tyrosine kinase inhibitors.

**Table 1 cancers-15-03015-t001:** Patient characteristics among each treatment cohort.

Characteristics	WBRT	SRS	WBRT + SRS	EGFR-TKIs + WBRT	EGFR-TKIs + SRS
Population (N)	32	51	36	29	18
Average age in years (N, range)	68.36 (38.91, 84. 95)	62.66 (38.42, 89.50)	64.76 (40.79, 89.88)	61.49 (43.66, 83.25)	70.27 (28.44, 90.34)
Female (%)	20 (62.5)	36 (70.6)	20 (55.6)	17 (58.6)	9 (50.0)
Multiple brain metastases (N, %)	20 (83.3)	27 (55.1)	20 (69.0)	23 (88.5)	15 (83.3)
Single brain metastases (N, %)	4 (16.7)	22 (44.9)	9 (31.0)	3 (11.5)	3 (16.7)
Extracranial metastases (N, %)	25 (80.6)	28 (54.9)	21 (60.0)	23 (82.1)	13 (72.2)
Leptomeningeal spread (N, %)	5 (16.7)	4 (8.3)	10 (29.4)	4 (14.3)	0 (0.0)
Symptomatic at time of brain metastases (N, %)	22 (81.5)	23 (53.5)	24 (70.6)	12 (60.0)	10 (58.8)
Type of EGFR-TKI					
Erlotinib/gefitinib				19 (65.5)	8 (44.4)
Osimertinib/afatinib				10 (34.5)	10 (55.6)

WBRT: whole-brain radiation therapy, SRS: stereotactic radiosurgery, EGFR: epithelial growth factor receptor, TKI: tyrosine kinase inhibitors.

**Table 2 cancers-15-03015-t002:** Survival statistics for each treatment cohort.

Cohort	Median OS (Months)	1-Year OS (95%CI)	2-Year OS Rate (95%CI)
WBRT only	3.23	35% (17%, 54%)	25% (10%, 44%)
SRS only	21.73	68% (51%, 80%)	39% (22%, 55%)
WBRT + SRS	3.17	26% (12%, 42%)	6% (1%, 19%)
EGFR-TKI + WBRT	15.50	64% (42%, 79%)	28% (12%, 46%)
EGFR-TKI + SRS	23.63	71% (43%, 87%)	37% (12%, 62%)

WBRT: whole-brain radiation therapy, SRS: stereotactic radiosurgery, EGFR: epithelial growth factor receptor, TKI: tyrosine kinase inhibitors, OS: overall survival.

**Table 3 cancers-15-03015-t003:** Multivariable analysis of survival using Cox’s proportional hazards model with adjustment for KPS, age, time from brain-metastases diagnosis to initiation of treatment, symptomatic brain metastases, and number of brain metastases.

Cohort	Hazard Ratio (95% CI)	*p*-Value
Treatment		
WBRT only	Reference	
SRS only	0.38 (0.17, 0.84)	0.017
WBRT + SRS	1.30 (0.60, 2.82)	0.50
EGFR-TKI + WBRT	0.93 (0.41, 2.08)	0.85
EGFR-TKI + SRS	0.46 (0.20, 1.09)	0.077
Age	1.02 (1.00, 1.04)	0.03
Time from brain metastases to treatment	1.00 (1.00, 1.00)	0.012
KPS		
Greater than 70	Reference	
Lesser than 70	2.49 (1.10, 5.66)	0.029
Symptomatic at time of brain metastases		
Asymptomatic	Reference	
Symptomatic	1.42 (0.89, 2.27)	0.14
Number of brain metastases		
Multiple	Reference	
Single	0.92 (0.53, 1.62)	0.78

WBRT: whole-brain radiation therapy, SRS: stereotactic radiosurgery, EGFR: epithelial growth factor receptor, TKI: tyrosine kinase inhibitors.

## Data Availability

The study lead authors, and the senior author have access to the primary dataset. Data may be made available to interested investigators upon reasonable request to the senior author (manmeeta@baptisthealth.net) after approval by all required regulatory authorities.

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
