# Peer review of "Combination of EGFR-Directed Tyrosine Kinase Inhibitors (EGFR-TKI) with Radiotherapy in Brain Metastases from Non-Small Cell Lung Cancer: A 2010–2019 Retrospective Cohort Study"

_cancers, 2023, doi:10.3390/cancers15113015_

Round 1

Reviewer 1 Report

1.This research focused on Combination of EGFR-directed Tyrosine Kinase Inhibitors (EGFR-TKI) with Radiotherapy in Brain Metastases from Non- Small Cell Lung Cancer: A 2010-2019 Retrospective Cohort

Study , after check the pubmed,although there are some references about EGFR-TKI Radiotherapy Brain Metastases lung cancer, this manusrcipt give so much clinical proof  of the benefit, very innovative.

2.But also some questions need to resolve:Firstly, in Table 1 the Patient Characteristics between groups  have some differences, how can you get the conclusion the OS or treatments have some differences?  As my opinion,

Compare 2 groups maybe ok such as WBRT and EGFR-TKIs + WBRT or SRS and EGFR-TKIs +SRS.

3.Ohter phase cohort  study also can be cited to Verify the effectiveness of your results such as PMID: 32634610.

4.I think the manusrcipt should cited the references between 2020-2023.

Author Response

We appreciate the reviewer's insightful comments. We believe the changes based on these comments and suggestions have improved the manuscript and strengthened our message.

We have responded to each of the issues raised by the reviewer in a point-by-point manner and are resubmitting the revised manuscript. Specific changes in the text have been called out in the responses in the attached file.

Reviewer 2 Report

The authors present a retrospective review of outcomes in EGFR mutated lung cancer cases with brain metastases treated with radiotherapy. They separate the treatment groups between WBRT vs SRS with or without TKI treatment.  SRS treatment with our without TKI shows favorable outcomes over the WBRT groups as the key finding.  While the article is of interest, I have several concerns/questions that should be addressed:

1. I find the descriptions of the treatment groups somewhat confusing, particularly those treated with TKI and those that were not.  Based on the nature of their metastatic disease, all patients should have been treated with TKI per standard of care.  The authors use a 28-day window for those treated concurrently (i.e. EGFR TKI-SRS group, EGFR TKI-WBRT group). Does that mean the other participants never received a TKI?  Clarification on this would be helpful in the methods section.

2. Number of brain metastases was excluded.  This may play a significant role in survival. inclusion of this data and subgroup analysis would be helpful to the findings.

3. There is no mention on whether brain metastases were symptomatic in each group. This data is also helpful on determining clinical outcome. 

4. As patients that were already on TKIs and progressed were included, the number of cases should be highlighted and they should also be analyzed as recurrent disease will have worsened survival over newly diagnosed cases. 

5. TKI treatment data per group should be included as some of the agents (erlotinib, gefitinib, etc have poorer blood brain barrier penetration). 

6. EGFR mutation data (particularly sensitive vs resistance mutation) and treatment outcome/group should be included, at a minimum as supplementary data, to understand if there were differences between groups. 

Author Response

(The authors gave the same response as above.)
